# Foot-and-Mouth Disease Virus 3C^pro^ Cleaves BP180 to Induce Blister Formation

**DOI:** 10.3390/v14092060

**Published:** 2022-09-16

**Authors:** Pathum Ekanayaka, Asela Weerawardhana, Kiramage Chathuranga, Jong-Hyeon Park, Jong-Soo Lee

**Affiliations:** 1College of Veterinary Medicine, Chungnam National University, Daejeon 34134, Korea; 2Animal and Plant Quarantine Agency, Gimcheon-si 39660, Korea

**Keywords:** FMDV 3C^pro^, BP180, FMDV 3C^pro^ C142T substitution, skin loosening, blister formation

## Abstract

Foot-and-mouth disease (FMD) is mainly characterized by blister formation (vesicles) in animals infected with foot-and-mouth disease virus (FMDV). However, the molecular basis of the blister formation in FMD is still unknown. BP180 is one of the main anchoring proteins connecting the dermal and epidermal layers of the skin. Previous studies have shown that the cleavage of BP180 by proteases produced by the inflammatory cells and the resulting skin loosening are major causes of the blister formation in bullous pemphigoid (BP) disease. Similar to BP, here we have demonstrated that, among the FMDV-encoded proteases, only FMDV 3C^pro^ contributes to the cleavage of BP180 at multiple sites, consequently inducing the degradation of BP180, leading to skin loosening. Additionally, we confirmed that FMDV 3C^pro^ interacts directly with BP180 and the FMDV 3C^pro^ C142T mutant, known to have reduced protease activity, is less effective for BP180 degradation than wild-type FMDV 3C^pro^. In conclusion, for the first time, our results demonstrate the function of FMDV 3C^pro^ on the connective-tissue protein BP180 associated with blister formation.

## 1. Introduction

Foot-and-mouth disease virus (FMDV), which belongs to the genus *Aphthovirus* within the family *Picornaviridae* [1], is a positive-sense RNA virus [2]. The FMD virus has unique characteristics enabling it to cause one of the most economically devastating diseases in cloven-hoofed animals, such as cattle and swine [3]. FMD is characterized by blister formation on the mouth, muzzle, tongue, snout, nose, teats, interdigital spaces of feet, and other hairless parts of the skin, causing off-feeding and lameness [4].

Like FMD, the autoimmune disease bullous pemphigoid (BP) includes blister formation as a major and common symptom. BP is characterized by the presence of autoantibodies against BP180 (collagen XVII) [5,6], which is a 180 kDa transmembrane glycoprotein. The primary role of BP180 is to act as a cell-matrix adhesion molecule that connects the dermal–epidermal layers [7]. The N-terminal region of BP180 is located in the hemidesmosomal plaque, while the C-terminal domain projects into the basal lamina of the skin [5]. In autoantibody-mediated inflammatory responses, proteases produced by the inflammatory cells induce BP180 cleavage, which has been suggested as a blister formation mechanism in BP [8]. In particular, inflammatory cells release proteases, such as mast cell protease (MCP)-4 and neutrophil elastase (NE), to cleave and release BP180, causing dermal–epidermal separation and blister formation [5].

The FMDV genome encodes the L^pro^, 3C^pro^, and 2A proteases that are important for polyprotein processing during viral replication [9]. Therefore, we hypothesize that, as in BP, the FMDV protease may be involved in the cleavage of BP180, forming blisters by loosening the skin through dermal–epidermal separation. In the present study, we suggest a possible molecular mechanism behind the blister formation in FMD via the FMDV-3C^pro^-mediated cleavage and degradation of BP180.

## 2. Materials and Methods

### 2.1. Plasmid Construction

The partial form of BP180 (1–600 aa) was purchased from the Korea Human Gene Bank, KRIBB (NM_000494; Clone ID: KU016286), and cloned into a pIRES vector tagged with V5 to generate pIRES_V5_BP180 plasmid. Gene–specific PCR primers were used to construct full-length, wild-type FMDV 3C^pro^, 3C^pro^ C142T point mutant, L^pro^, and 2A+2B genes of the O1/Manisa/Turkey/69 strain and the PCR products were cloned into pEXPR-Strep, pIRES-FLAG, or pEBG-GST expression vectors to generate pEXPR_Strep_3C^pro^-WT, pEXPR_Strep_3C^pro^-C142T, pIRES_FLAG_L^pro^, and pEBG_GST_2A+2B plasmids, respectively.

### 2.2. Cell Culture and Plasmid Transfection

HEK293T cells were cultured in Dulbecco’s Modified Eagle’s Medium (DMEM-high glucose; Gibco, Thermo Fisher Scientific, Waltham, MA, USA) containing 10% heat-inactivated fetal bovine serum (FBS; Gibco) and 1% antibiotic/antimycotic solution (Gibco). Cells were incubated at 37 °C in a 5% CO_2_ atmosphere. Plasmids were transfected into HEK293T cells using a polyethylenimine (PEI) reagent.

### 2.3. Antibodies and Inhibitors

For immunoblot analysis, antibodies against V5 (A190-220A) were purchased from BETHYL Laboratories (Montgomery, TX, USA) and Strep (2-1509-001) from IBA Life Sciences (Gottingen, Germany). Anti-FLAG antibody (M2) (8146) was purchased from Cell Signaling Technology (Danvers, MA, USA). For the detection of β-actin (sc-47778) protein and GST (sc-138), antibodies were purchased from Santa Cruz Biotechnology. The inhibitors MG132 (M8699), chloroquine (CQ) (C6628), ammonium chloride (A9434), and rupintrivir (PZ0315) were purchased from Sigma-Aldrich (St. Louis, MO, USA), and Z-VAD-FMK (sc-3067) was purchased from Santa Cruz Biotechnology (Dallas, TX, USA).

### 2.4. Strep Pull-Down Assay

The target plasmids were co-transfected into the HEK293T cells, and 30 h post-transfection, cells were harvested. To generate whole-cell lysates (WCLs), cells were lysed with radio-immunoprecipitation assay (RIPA) lysis buffer (50 mM Tris-HCL, 150 mM NaCl, 0.5% sodium deoxycholate, 1% IGEPAL, 1 mM NaF, and 1 mM Na_3_VO_4_) containing a protease and phosphatase inhibitor cocktail (Sigma). WCLs were incubated with Sepharose 6B resin (GE Healthcare Life Science, Chicago, IL, USA) at 4 °C for 2 h for pre-clearing, followed by the Strep pull-down. For the Strep pull-down, pre-cleared WCLs were incubated for 12 h with a 50% slurry of Strep-Tactin Sepharose Strep beads (IBA Solutions for Life Sciences, Gottingen, Germany). Immunoprecipitated beads were collected by centrifugation and washed with lysis buffer for immunoblot analysis.

### 2.5. Immunoblot Analysis

The cells were lysed with RIPA lysis buffer in the presence of the protease and phosphatase inhibitor cocktail (Sigma). Individual proteins in the samples were separated via SDS-PAGE and transferred onto a PVDE membrane (Bio-Rad, Hercules, CA, USA) using a Trans-Blot semi-dry transfer cell (Bio-Rad) with buffer containing 30 mM Tris, 200 mM glycine, and 20% methanol. The transferred membranes were incubated at room temperature for 1 h with 5% bovine serum albumin (BSA) in 1× TBST (Tris-buffered saline containing 0.05% Tween 20) for blocking. Then, the membranes were incubated with the target primary antibody at 4 °C overnight, and following that, they were washed 3 times with 1× PBST or TBST for 10 min each. After this step, the membranes were treated with horseradish peroxidase (HRP)-conjugated secondary antibody for 1 h at room temperature, and then, the washing step was repeated another 3 times. The HRP was visualized using an Enhanced Chemiluminescence Femto kit (LPS solution, Daejeon, Republic of Korea) and a LAS-4000 Mini Lumino Image Analyzer (GE Healthcare Life Sciences, Chicago, IL, USA).

## 3. Results

### 3.1. The FMDV 3C^pro^ Protease Cleaves and Degrades BP180

Blister formation is a major symptom of FMDV infection [4]**,** and BP180 cleavage by host proteases is known to form blisters in autoimmune diseases [5,8]. FMDV encodes several proteases important for polyprotein processing during viral replication [9]. Therefore, we first evaluated the effect of FMDV proteases on BP180.

BP180 is a 180 kDa protein, which is technically difficult to manipulate for biochemical experiments. Therefore, we used the truncated BP180 protein consisting of the first 600 aa (hereafter referred to as BP180 (1–600 aa)) to alleviate the complications of handling a large protein. BP180 (1–600 aa) is 66 kDa in size and consists of the BP180 intracellular domain and part of the extracellular domain containing the major immunogenic region NC16A (Figure 1A). HEK293T cells were cotransfected with C-terminal V5-tagged BP180 (1–600 aa) together with increasing amounts of Strep-tagged wild-type FMDV 3C^pro^, FLAG-tagged FMDV L^pro^, or GST-tagged FMDV 2A+2B, and immunoblotting was performed to evaluate the effect of FMDV proteases on BP180. As shown in Figure 1B–1D, only wild-type FMDV 3C^pro^ cleaved and degraded BP180 in a dose-dependent manner. Since we used C-terminal V5-tagged BP180 (1–600 aa), based on Figure 1B, wild-type FMDV 3C^pro^ was shown to cleave BP180 at four main cleavage sites into four main-cleaved BP180 fragments representing molecular weights of 60, 55, 30, and 20 kDa, and BP180 was sensitive enough to be degraded even at very low 3C^pro^ doses. Collectively, these results suggest that BP180 is specifically cleaved and degraded by wild-type FMDV 3C^pro^ but not by other FMDV-encoded proteases.

### 3.2. The Protease Activity of FMDV 3C^pro^ Governs Its Ability to Degrade BP180

To determine whether wild-type FMDV 3C^pro^ affects the specific type of degradation of BP180, we cotransfected the Strep-tagged wild-type FMDV 3C^pro^ plasmid and the V5-tagged BP180 (1–600 aa) plasmid into HEK293T cells treated with different inhibitors, including the proteasomal inhibitor MG132; the lysosomal inhibitors CQ or NH_4_Cl; the pan-caspase inhibitor Z-VAD; and a broad-spectrum protease-activity inhibitor, rupintrivir [10]. In a previous study, rupintrivir was shown to be a picornavirus-specific 3C^pro^ protease activity inhibitor [11]. As shown in Figure 2, BP180 expression levels were restored upon treatment with rupintrivir, but not with MG132, CQ, NH_4_Cl, or Z-VAD. These results indicate that BP180 undergoes protease-mediated cleavage and degradation by wild-type FMDV 3C^pro^, but not by proteasomes, lysosomes, or caspases.

### 3.3. FMDV 3C^pro^ Interacts with and Degrades BP180

As a result of the aforementioned tests, it was confirmed that the protease activity of FMDV 3C^pro^ governs the cleavage and degradation of BP180. However, we further validated whether FMDV 3C^pro^ protease activity is required for the degradation of BP180. The C142T mutation of FMDV 3C^pro^ significantly decreases protease activity [12] and moderately attenuates FMDV pathogenicity [11]. Therefore, the ability of FMDV 3C^pro^ C142T to degrade BP180 was compared with that of wild-type FMDV 3C^pro^ as another method for validation. To investigate the effect of FMDV 3C^pro^ C142T on BP180 degradation, HEK293T cells were co-transfected with Strep-tagged wild-type FMDV 3C^pro^ or FMDV 3C^pro^ C142T plasmid in a dose-dependent manner, or with control plasmid together with V5-tagged BP180 (1–600 aa) plasmid. Cell lysates were then immunoblotted with V5, Strep, or β-actin antibodies. As shown in Figure 3A, BP180 was susceptible to cleavage and degradation by both wild-type FMDV 3C^pro^ and 3C^pro^ C142T. However, the ability of FMDV 3C^pro^ C142T to degrade BP180 was lower than that of wild-type FMDV 3C^pro^ due to reduced 3C^pro^ C142T protease activity. These results again indicate that BP180 is highly sensitive to FMDV 3C^pro^ protease activity.

Additionally, to determine whether FMDV 3C^pro^ interacts with BP180 for the cleavage, we performed an immunoprecipitation assay using wild-type FMDV 3C^pro^ and FMDV 3C^pro^ C142T. HEK293T cells were separately transfected with V5-tagged BP180 (1–600 aa) plasmid, control plasmid, Strep-tagged wild-type FMDV 3C^pro^, or FMDV 3C^pro^ C142T plasmid, and then they were treated with rupintrivir.

As shown in Figure 3B, wild-type FMDV 3C^pro^ and FMDV 3C^pro^ C142T interacted with BP180 (1–600 aa) despite differences in protease activity. Taken together, these results suggest that FMDV 3C^pro^ specifically interacts with BP180 and mediates the cleavage and degradation of BP180 through 3C^pro^ protease activity.

## 4. Discussion

The skin is a multifunctional organ that protects the body from pathogens and harmful environments [13,14]. The epidermis, the outermost layer of the skin, contains squamous and stratified epithelium and is mainly composed of keratinocytes. On the other hand, the dermis serves as the middle layer of the skin and consists of an integrated system of fibrous, filamentous, and connective tissues, as well as nerve and vascular networks [14]. BP180 is one of the major proteins anchoring the dermis to the epidermis at their junction [15]. Skin-loosening through cleavage and degradation of BP180 by inflammatory cell-driven proteases is a key molecular mechanism of blister formation in the autoimmune disease bullous pemphigoid (BP) [5,8]. As with BP, blister formation is one of the main symptoms of foot-and-mouth disease (FMD) [4], and FMDV produces several proteases important for viral polyprotein processing during viral replication [2,9]. However, the molecular mechanisms and involvement of viral proteases in FMDV-induced blister formation are still unknown.

In the present study, the molecular mechanism behind blister formation in FMD was elucidated based on several observations. First, we showed that, among the FMDV-encoded proteases, FMDV 3C^pro^ is the only one involved in the cleavage and degradation of BP180. Second, we demonstrated that the protease activity of FMDV 3C^pro^ governs the degradation of BP180 based on the use of rupintrivir, a broad-spectrum picornavirus 3C protease activity inhibitor [10]. Finally, we confirmed that FMDV 3C^pro^ specifically interacts with BP180 and the 3C^pro^ protease activity-mediated cleavage of BP180 using the FMDV 3C^pro^ C142T mutant.

FMDV 3C^pro^ is a chymotrypsin-like serine protease playing a crucial role in FMDV polyprotein processing [9,12]. Additionally, FMDV 3C^pro^ inhibits host cellular processes through the cleavage and degradation of host proteins. For example, FMDV 3C^pro^ is known to block host protein synthesis by cleaving eIF4AI and eIF4G translation initiation factors [16,17] and to inhibit the nuclear factor kappa B (NF-κB) pathway by cleaving NF-kB essential modulator (NEMO) [18]. Additionally, the protease activity of FMDV 3C^pro^ is responsible for the degradation of the autophagy-related 5-12 (ATG5-ATG12) protein complex [19], karyopherin subunit alpha 1 (KPNA1) [20], retinoic acid-inducible gene I (RIG-I), melanoma differentiated assisted protein 5 (MDA5) [11,18] and dsRNA-activated protein kinase R (PKR) [21] to negatively regulate host antiviral responses.

In this study, we demonstrated that the protease activity of FMDV 3C^pro^ induces the degradation of BP180 using rupintrivir and FMDV 3C^pro^ C142T plasmid. Amino acid residue 142, located at the tip of the FMDV 3C^pro^ β-ribbon, is critical for optimal enzymatic activity and substrate recognition [12]. The C142S substitution almost completely inhibits FMDV 3C^pro^ protease activity, leading to the complete inactivation of the virus, whereas the C142T substitution reduces the enzymatic activity of wild-type FMDV 3C^pro^ by more than 60% [12]. Therefore, Figure 3A shows that the cleavage and degradation of BP180 were reduced by the residual enzymatic activity of FMDV 3C^pro^ C142T.

In previous studies, FMDV 3C^pro^ has been shown to cleave eIF4AI and eIF4G translation initiation factors to inhibit host protein synthesis [16,17]. However, our results show that the main cause of BP180 degradation by FMDV 3C^pro^ is the direct cleavage of BP180 at multiple sites, such as the targeting of RIG-I and MDA5 [11], instead of the FMDV 3C^pro^-mediated inhibition of host protein synthesis.

Previous studies have shown that mouse BP180 is cleaved by the host protease, neutrophil elastase (NE), resulting in dermal–epidermal layer separation and eventual skin loosening [22]. After cleavage of BP180, the cleaved BP180 fragments, acting as a chemotactic factor for neutrophils to induce neutrophil infiltration into the BP180 cleaved site, trigger an inflammatory response. The inflammation then appears as blisters due to the accumulation of fluid between the loose layers of the skin [22]. In this study, we have assumed that BP180 cleavage by the FMDV 3C^pro^ might also induce blister formation in FMD via the above-described mechanism. However, further studies are required to prove it.

In summary, for the first time, this study shows that, among the FMDV-encoded proteases, FMDV 3C^pro^ specifically interacts with BP180, a key anchoring molecule linking the dermal–epidermal layers, and that the protease activity of FMDV 3C^pro^ governs the cleavage and degradation of BP180. Since BP180 is a key molecule associated with blister formation, these findings may stimulate further studies leading to an understanding of the molecular mechanism of blister formation in FMD for the development of a rational approach for future FMD prevention strategies.

## Figures and Tables

**Figure 1 viruses-14-02060-f001:**
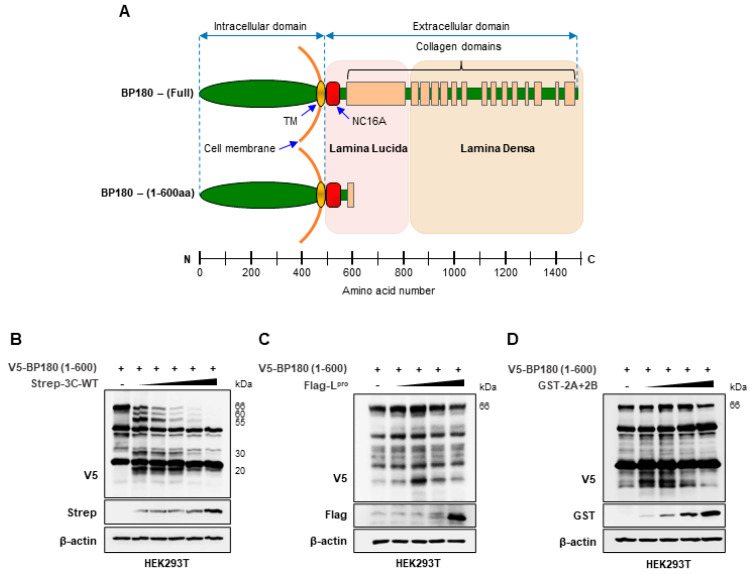
FMDV 3C^pro^ cleaves and degrades BP180: (**A**) Diagram illustrating the domain mapping of BP180; (**B**–**D**) Respective immunoblotting of BP180 cleavage and degradation with FMDV proteases. The HEK293T cells were cotransfected with V5–tagged BP180 (1–600 aa) plasmid (1.5 µg) along with the control plasmid or increasing amounts of (**B**) Strep–tagged FMDV 3C^pro^ wild–type plasmid (10, 20, 40, 80, and 160 ng), (**C**) FLAG–tagged FMDV L^pro^ plasmid (100, 200, 400, and 800 ng), or (**D**) GST–tagged FMDV 2A+2B plasmids (100, 200, 400, and 800 ng) on a 6–well plate. Samples were collected at 30 h post–transfection and cell lysates were immunoblotted with V5, Strep, FLAG, GST, and β–actin antibodies based on the tagging of the transfected plasmids. All of the data represent 3 independent experiments, each with similar results.

**Figure 2 viruses-14-02060-f002:**
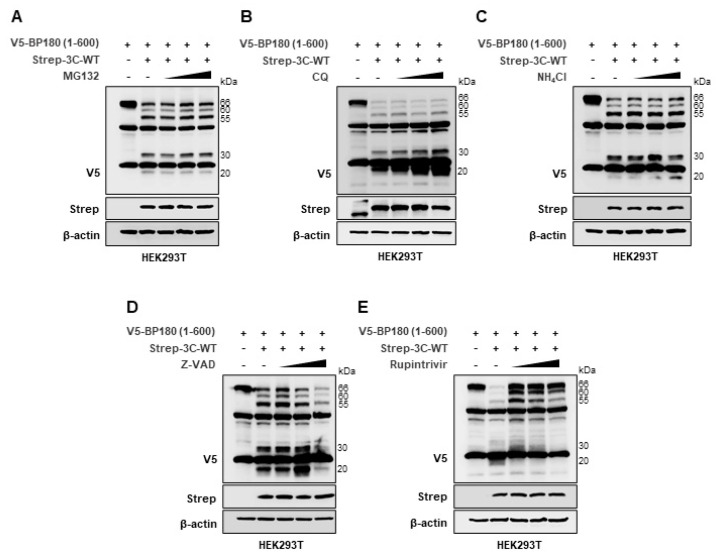
FMDV 3C^pro^ degrades BP180 through its protease activity. HEK293T cells were transiently transfected with V5–tagged BP180 (1–600 aa) plasmid (1.5 µg) together with control plasmid or Strep–tagged FMDV 3C^pro^ wild–type plasmid (80 ng), and treated with (**A**) MG132 (5, 10, 20 µM), (**B**) CQ (50, 100, 200 µM), (**C**) NH_4_Cl (10, 20, 50 mM), (**D**) Z–VAD (10, 50, 100 µM), or (**E**) Rupintrivir (10, 20, 40 µM). The HEK293T cells were treated with MG132 for 8 h before the samples were collected, and the cells were treated with all other inhibitors for 18 h before the samples were collected. Samples were collected at 30 h post-transfection, and cell lysates were immunoblotted against the V5, Strep, and β–actin antibodies. All of the Western blot data represent 3 independent experiments, each with similar results.

**Figure 3 viruses-14-02060-f003:**
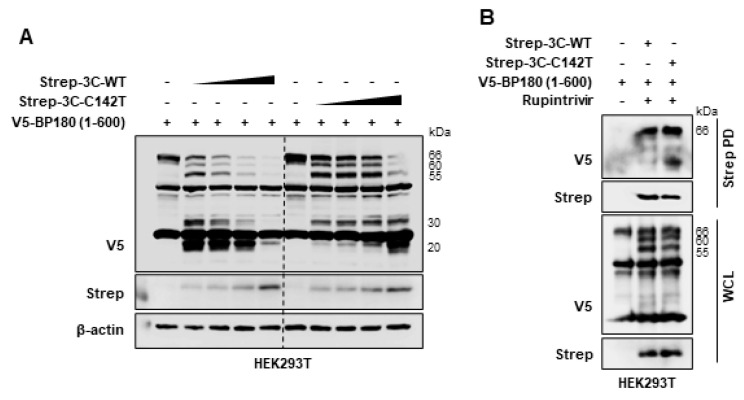
FMDV 3C^pro^ interacts with BP180: (**A**) HEK293T cells were transiently transfected with Strep–tagged wild–type FMDV 3C^pro^ or 3C^pro^ C142T plasmids in a dose–dependent manner (10, 20, 40, and 80 ng) or control plasmid, together with V5–tagged BP180 (1–600 aa) plasmid (1.5 µg) in 6–well plates. Samples were collected 30 h post–transfection, and cell lysates were subjected to immunoblotting with V5, Strep, and β–actin antibodies; (**B**) HEK293T cells were co-transfected with a V5–tagged BP180 (1–600 aa) plasmid (2 µg), together with a control plasmid, Strep–tagged wild-type FMDV 3C^pro^, or 3C^pro^ C142T plasmid (0.5 µg), and subsequently treated with rupintrivir. Samples were collected at 30 h post–transfection, and cell lysates were subjected to Strep pull–down (StrepPD) and immunoblotted with antibodies against V5, Strep, and β–actin. All of the data represent at least 2 independent experiments, each with similar results. (WCL—whole cell lysate).

## Data Availability

All relevant data are contained within the manuscript.

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
