# Peer review of "Foot-and-Mouth Disease Virus 3C^pro^ Cleaves BP180 to Induce Blister Formation"

_viruses, 2022, doi:10.3390/v14092060_

Round 1

Reviewer 1 Report

Comment on Ekanayaka et al.

This manuscript describes the ability of 3Cpro, the main protease, of foot and mouth disease virus (FMDV) to cleave BP180, the cellular protein associated with the blister formation in an autoimmune disease called Bullous pemphigoid (BP). The authors demonstrated that among FMDV proteases including Lpro, 3Cpro, and a self-cutting 2A, only 3Cpro could cleave the BP180 at multiple positions. The cleavage of BP180 by 3Cpro was shown to be dose-dependent. The 3Cpro mutant, C142T with the reduced protease activity, could cleave the BP180 but less effective. In addition, a protease inhibitor, rupintrivir, could protect BP180 from the 3Cpro cleaving. Although 3Cpro has been shown to cut several host proteins involved in interferon induction and response pathway, this is the first report demonstrating the function of 3Cpro on the connective tissue protein, BP180. However, the evidence from this study is not sufficient to conclude that the blister lesion is caused by 3Cpro.

Major concern

1.       To conclude that 3Cpro caused the blister formation, the authors must show co-localization of 3Cpro and BP180 at the junction between dermis and epidermis in the blister lesions of FMDV infected animals or organ culture. Therefore, the title should be changed to a more appropriate one such as “Foot and mouth disease virus 3Cpro cleaves BP180, the protein associated with blister formation”. Also, the conclusion should be revised to avoid overstatement.

2.       In the abstract, the authors quite overstated that their finding “provided information for the development of live attenuated FMFV vaccine”.  Their study only showed that 3Cpro could cleave BP180 and C142T reducing the protease activity of 3Cpro. Therefore, the phrase “and could be provided for the development of live attenuated FMDV vaccine” (line 20) should be removed. Indeed, 3Cpro was shown to be a multipurpose protease that cleave several cellular proteins associated with inhibition of the antiviral cytokine, IFN. In addition, the authors linked it with the blister formation of FMDV, and 3Cpro with C142T reduced the protease activity. It would be interesting if the authors elaborate on their ideas in the discussion of how their findings could lead to the development of a live attenuated vaccine.

3.       In Materials and Methods; 2.1 (Lines 49-54), the authors should indicate the resulting plasmid names after cloning. There were 4 plasmids used in this study, pIRES_V5_BP180, pEXPR-Strep_3Cpro, pIRES_Flag_Lpro, and pEBG_GST_2A-2B.

4.       Results; 3.1, the authors should provide the detail of BP180 cleavage by 3Cpro. What is the molecular weight (MW) of BP180 (1-600 aa)? How many cutting sites are there in the BP180? The authors may analyze the potential 3Cpro cleavage sites on the BP180 amino acid sequence and compare them with the in vitro experiment. It will be clearer if a schematic diagram of the 3Cpro cleavage sites with their MWs on the BP180 will be drawn and put beside Figure 1A.  What are the MWs of the cut BP180? The authors should indicate the MWs of the important protein bands in kDa in all Figures.  

5.       In Figures 1B-D and 3A, the plasmids were transfected in HEK293 cells at different amounts to show the dose-dependent effects of 3Cpro, Lpro, 2A-2B, and 3Cpro C142T. The authors should indicate µg of each plasmid used in each transfection. In addition, all plasmids used for transfections in every experiment should be indicated in all Figueres.

6.       Figure 3B; Indicate µg of all proteins used in immunoprecipitation. The abbreviations presented beside the picture (StrepPD and WCL) should be explained in the Figure caption.

7.       The manuscript must be revised by an English professional service for scientific publication.

Author Response

Dear Reviewer,

We would like to thank the reviewer for evaluating our work, for insightful comments, and for the valuable advice. We thoughtfully considered the comments and revised the manuscript following the suggestions, and now we believe the manuscript is greatly improved.

Thank you.

Reviewer 2 Report

The study by Ekanayaka et al. seems to be very interesting, but there are some major concerns the authors must address before it could be considered for publication.

Why do the authors use only 600 amino acids sequence of BP180 (composed of intracellular and extracellular domains)? It is a very long protein. How many domains are there in total in the BP180?

Figure 1 showed that 3Cpro, Lpro, or 2A+2B were used in a dose-dependent manner, but the authors did not mention how much dose of each of the FMDV 3Cpro or Lpro or 2A+2B were used. I suggest showing the dose in the figure legend or main text and after how many hours the samples were collected. Besides, the authors should show the results of the cell viability assay of Figures 1B, C, and D.

We have seen in figure 1 that the use of FMDV protease, 3Cpro, degradation and cleavage of BP180 was observed. I suggest the authors should use the increasing concentrations of BP180 and see its degradation effect on FMDV protease, 3Cpro.

Why do the authors use HEK293T cells instead of PK-15 or BHK-21 cells? Can the authors describe the detailed reasons for using HEK293T cells?

Figure 2 showed the use of proteasome, lysosome, caspase, and Rupintrivir inhibitors, but the authors did not mention after how many hours these inhibitors were used. And after how many hours post-transfection were the samples collected?

L142 “However, we one more validated” what does it mean? Please modify the sentence.

In figure 3, the authors used Rupintrivir in both wild-type and protease mutant transfected samples. Will it not affect the binding activity between BP180 and FMDV 3Cpro and mutant? It is strongly recommended that the authors use wild-type 3Cpro and BP180 transfection assay without using Rupintrivir and subsequent CO-IP. 

Author Response

Dear Reviewer,

We thank the reviewer for evaluating our work, and for the valuable advice. We did our best to follow the reviewers’ comments to improve the quality of our manuscript.

Thank you
